# Romanian Holiday Vouchers: A Chance to Travel for Low-Income Employees or an Instrument to Boost the Tourism Industry?

Claudia Daniela Albă 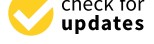 and Liliana Sonia Popescu *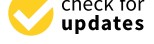

Department of Geography, University of Craiova, 200585 Craiova, Romania
* Correspondence: liliana.popescu@edu.ucv.ro

**Abstract:** Holiday vouchers are a tool that have been used for 40 years to encourage employees' access to vacation and have been highlighted during the recent pandemic, when governments used them to support the tourism industry. Using a naturalistic inquiry that combined focus groups with workers from travel agencies and semi-structured interviews with tourists, we analysed the influence of the Romanian holiday voucher scheme on the tourist behaviour of employees in order to establish the holiday vouchers' usefulness as a tool in social tourism or for the development of the tourism industry. An overwhelming share of the interviewees considered the granting of holiday vouchers beneficial. First and foremost, holiday vouchers enabled some people with blue-collar jobs to have their first holiday in decades. Secondly, vouchers influenced tourist behaviour mainly in terms of destination choice, services purchased and holiday frequency and/or duration. Moreover, for the past couple of years when there were major events with impacts on the entire economy worldwide, holiday vouchers proved to be a lifesaver for the Romanian tourism industry.

**Keywords:** holiday vouchers; tourist behaviour; social tourism; semi-structured interviews; focus group; Romania

## 1. Introduction

Legislation and political decisions have a significant impact on the touristic phenomenon [1], and the decision to grant some holiday vouchers as well as the institutionalisation of companies reimbursing employees' holidays can be an intervention lever in the tourism industry. Moreover, in the post-pandemic stage, countries are paying attention to implementing strategic measures in order to stimulate tourism demand [2], such as granting holiday vouchers. Social tourism developed in the 20th century to include ideas of social inclusion and welfare rights [3], which is why it has been described as an umbrella concept [4] to refer to its complexity, targeting economically weak or otherwise disadvantaged people. The basic principle is to enable access to travel and leisure opportunities for all [5], referring to the goal to include as many people as possible in tourism and leisure [6].

Although holiday voucher schemes started to be used in Romania in 2015, there is little research regarding the implementation and the results of using this means of purchasing vacations. The few studies about Romanian holiday vouchers only analyse the official statistics related to general tourist flows, influenced or not by the vouchers [7,8], or some estimations made by the National Association of Travel Agencies in Romania about the number of issued vouchers and some of the holders' choices [9]. Another study presents the main reasons for traveling and the positive effects highlighted by employers, tour operators and travel agencies [10]. There is no study focusing on the impact that the implementation of the voucher system had on various categories of employees or the changes in the tourist behaviour of people who used to travel quite frequently.

Therefore, the current study aims at analysing the influence of holiday vouchers on the behaviour of tourists who were accustomed to spending their holidays in Romania,

as well as those who preferred holidays abroad and also those who had never been on a holiday before, thus approaching a social tourism component. The willingness to accept the holiday vouchers as a means of payment by the travel agencies is also analysed.

Thus, this study brings an important contribution to the understanding of the holiday voucher system (application, benefits, downsides, things that must be improved). The study is a useful tool for the Romanian authorities with this initiative, carrying out an evaluation of the application of the holiday voucher system and highlighting the benefits brought to a social category that was not specifically targeted by this initiative.

Moreover, this model could be applied to other countries, especially those that emphasise the development of the tourism sector. The use of the holiday voucher system as a tool to increase social inclusion is an important benefit in the field of social tourism.

The paper is structured as follows. We first provide an image of the concepts approached, as well as the conceptual framework. Then, the key questions are presented, and Section 5 describes the research design. Results are discussed in Section 6, focusing on findings following the focus groups and tourists' perception based on semi-structured interviews. Implications for destination managers are also highlighted. Section 7 concludes the study.

## 2. International Background and Concepts Approached

### 2.1. The Concept of the Holiday Vouchers

Most commonly, the term "voucher" is understood as an instrument that allows the holders to benefit from some discounts [11]; it can also be redeemed for a service, commodity or other such benefit provided by an agent [12]. Social vouchers generally facilitate the access of some social categories to particular services or products [13]. The main characteristics of social vouchers are that they are regulated by a legal framework, have a limited geographical scope, are provided upon request of a private or public entity, offer access to particular service providers, should be easy to use and cannot be exchanged for money [14,15]. According to the Social Vouchers International Association [16], which was established in Belgium in 2017, social vouchers have been developed in 40 countries, 19 of which are EU countries, over the past 50 years and include the following categories: (1) food and meal vouchers, (2) personal and household services vouchers, (3) transport vouchers, (4) leisure vouchers, (5) childcare vouchers, (6) culture vouchers and (7) eco-vouchers. The purpose of leisure vouchers is to facilitate citizens' access to a healthy lifestyle, and sport and wellness facilities.

The concept of holiday vouchers was introduced in France in 1982 with "cheque vacances", an instrument aimed at facilitating the access of employees to holidays, and with the establishment of a national agency for chèques-vacances [17]. The vouchers were valid for the year when they were issued until 31 December the next year; those that were not used could be redeemed for cash during the first 3 months after their validity. However, France was not the first country to facilitate employees' access to holidays. In 1939, the Swiss government established the Swiss Travel Saving Fund (REKA), an institution aimed at encouraging travel and holidays for people with limited financial means [18]. The chèques-vacances system is still applied in France, being considered an instrument for fighting against social exclusion. These vouchers can be used for travelling, hotels, youth hotels, catering, cultural activities and athletic activities [19]. They are distributed based on social criteria to employees from private companies, public employees as well as to people who work independently [20].

Before the pandemic, the tourism industry reported a continuous expansion, becoming one of the sectors with the fastest growth worldwide [21], with the 2010–2019 period being seen as the Golden Age for the European tourism industry [22]. However, economic conditions, political decisions and social changes transform the tourism environment and increase competition [23]. In order to support domestic tourism, some European countries provided the regulatory legal framework for the reimbursement of employees' holidays by employers or for granting holiday vouchers. Beginning on 1 January 2019, Slovakia introduced an amendment to Act No. 91/2010, urging employers with more

than 49 employees to subsidise 55% of the cost of the holidays for those working for the company for at least 1 full year. "Recreational contribution" paid by the employer could be up to EUR 275 per calendar year [24]. Spain or Lithuania have also implemented a programme for vouchers for travel or recreation to be used by children or families with children, having special needs or limited financial activities [19], with these countries focusing on social tourism.

Over the past two years, more countries or particular regions within a country have turned to tourism vouchers under one form or another (staycation vouchers/travel subsidies) to sustain inbound tourism. Due to restrictions generated by the COVID-19 pandemic, the tourism industry was one of the hardest hit economic sectors [25–30]. The introduction of social vouchers in an attempt to boost domestic travel was justified by a sharp decline in tourism demand, loss of jobs and closure of non-essential industries [31]; this measure was reported in countries such as Island, Japan, Slovenia, Ireland, Italy, Poland, South Korea, Taiwan, China, etc. [31–34].

### 2.2. Social Tourism and Social Inclusion

Initially, the concept of social tourism, strongly rooted in the ideology of social democratic traditions in France [35], was described as the participation in travel by people with a low income [36], referring to budget-friendly holidays in their own country that are funded totally or partially by charities or agencies in the public sector [37]. Currently, the focus of social tourism has shifted considerably towards social inclusion and cohesion [4,38], especially within Europe, where both Western and Eastern countries have fostered the idea of social tourism as an obligation a state owes its citizenry and its society in order to fulfil the right to tourism espoused in charters such as the Universal Declaration of Human Rights [39]. This is why social tourism has become a flagship tourism policy in the European Union, since it best represents the ideals, aims and objectives of a truly social Europe [40].

The social tourism aims for the inclusion of a particular category of users who otherwise would be excluded. These beneficiaries wish to travel, but they cannot afford it [41]. According to the official statistics of the European Union, between a quarter and a third of the EU population aged 16 and over could not afford a one-week annual holiday away from home during the past decade, although the share is continuously decreasing (from 39.5% in 2013 to 28.6% in 2020). Among the member states, this greatly varies from as low as only 10% in Sweden to more than 50%. Romania ranks first among the countries with the highest proportion of individuals in this situation: 66.6% in 2013, 58.9% in 2018 and 54.1% in 2019 [42].

The benefits of social tourism are twofold: firstly, there are clear social benefits regarding the access of disadvantaged members of society to travel opportunities; secondly, social tourism is linked with tourism sustainability, especially in the areas with high dependencies on tourism.

The core value of social tourism stems from the idea that "having a break from daily life (and problems) contributes to good health" [43]. Research results support the idea that holidays play an important role in ensuring equality for disadvantage people, providing them a sense of feeling included in society [38,44]. Apart from the need for social inclusion, recent findings demonstrate various psychological benefits such as increased self-esteem and positive changes in job research-related behaviour [45], quality of life [46] and well-being [44,47–49], overall functioning [48] and family and social capital [49,50]. Moreover, using qualitative data, Kakoudakis et al. [45] highlighted the fundamental role played by the holiday environment, through creating enabling conditions and safe spaces, triggering positive cognitive and behavioural changes. Considering that social tourism can have positive outcomes not only for the tourists, but also for the social and welfare policy and society as a whole, the needs of people in developing countries must be assessed to ensure that policies and programmes for social tourism are beneficial and relevant to their needs [35]. This assessment is highly necessary, as for any policy, there must be a basis in "evidence" [43].

Recently, there was a shift in the rationale of social tourism, focusing not only on the demand-side perspective, but also on the economic benefits of this type of tourism, since

it has the potential to sustain the inflow of tourists during the low season [37,51,52], thus sustaining jobs and generating income for host communities.

*2.3. Tourist Behaviour*

Starting from the grouping of the factors that influence tourist behaviour in pull and push factors [53], then the theory of planned behaviour [54] or the models that explain tourists' choices [55–60], consumer behaviour is considered one of the most researched areas in the field of tourism [61,62]. Cohen et al. reviewed more than 500 articles published in three major tourism journals, establishing nine key concepts for tourism consumer behaviour: decision making, values, motivations, self-concept and personality, expectations, attitudes, perceptions, satisfaction, and trust and loyalty [62].

Decision making, understood as the way to define consumers and their behaviour [63], is based on a negotiation process between tourists' needs and amenities offered by destinations [64]. According to decision-making styles, Decrop and Snelders identified six type of tourists: habitual, rational, hedonic, opportunistic, constrained and adaptable. Habitual tourists repeat the same vacation behaviour almost every year; the rational tourist is not a daydreamer like the hedonic vacationer, but a careful and realistic decision maker with well-defined decision criteria and strategies; the opportunistic tourist does not use any well-defined strategy and the decisions result from chances; constrained tourists are weighed down by contextual inhibitors such as limited financial resources or the intervention of situational variables and are not really involved in a decision-making process; adaptable tourists conform their plans according to the situation, which means that they often revise their decisions and modify their behaviour [65]. Another classification according to decision-making styles includes the following categories of tourists: perfectionist or high-quality consciousness; brand consciousness; price and value for money; confusion by over-choice; habitual, brand-loyal orientation.

The choice of the holiday destination involves multiple decisions [62,66,67] and is influenced by various factors [68] such as physical attributes, attractions or intra-attractions [69–71], the previous satisfaction regarding a destination [72,73], tourists' expectation [74,75] or the destination image [76], the distance from home to destination [77,78] and travel mode [79], interactions between tourists [80–82] or reciprocal resident–tourist relationship [83], the prestige of being in a place or ego enhancement [84], and above all, the tourist expenditure is a basic component of tourism demand [85,86].

The substantial influence that price has on the decision to choose vacations was confirmed in some studies carried out in Slovakia [87] and Romania [88]. In periods of economic recession or crises, the tendency of economising or changing the travel planning strategy was observed in the decision to choose vacations [89], and granting some facilities or promotions are critical in making the consumer's purchase decision [90].

## 3. Study Context: Romania—Conceptual Framework

The concept of holiday vouchers was first used in Romania in 2009, when Ordinance no. 8/2009 was approved; however, it became effective only after 2015, after several other laws were published [91]. Holiday vouchers can be granted to all employees, regardless of their wage or costs being covered by the employers, either public or private entities; moreover, they must be used only for tourism packages including at least accommodation for one night at a certified accommodation facility in Romania.

There were multiple aims of the voucher scheme, namely recovery and support of the employees capacity to work [92], diminishing undeclared income by forcing accommodation facilities to obtain proper certification [93], enticing the workforce to not leave the country and preventing the exodus of the tourism staff [94] and, later on, mitigation of the negative effects of the COVID-19 pandemic on the tourism sector [95]. All employees can receive vouchers, and only holidays in Romania are allowed; they can be purchased directly from authorised and certified accommodation units or from licensed travel agencies. Affiliated units that can accept vouchers must not have any financial dues to the state

budget. Holiday vouchers can be granted by both public and private units but they are an optional benefit, with employers deciding for or against it depending on the company budget or the collective/individual employment contract. For the employer, the advantage of vouchers instead of bonuses for the paid leave stems from the fact that there are no contributions to be paid to the state budget for them.

The value of holiday vouchers granted for public employees beginning on 1 January 2019 is around EUR 300 per calendar year [95], which is roughly the equivalent of the minimum monthly net wage in Romania, while for employees of private companies, the maximum amount is the equivalent of six gross minimum wages/year, and the same amount should be provided for each and every employee, no matter the position within the company. Still, most of the vouchers were granted by public and not private entities. No matter if it is public or private entity, it must cover the costs for the holiday vouchers, and the employees only pay a 10% tax. Another limit is set for the agencies' profit margin, which cannot exceed 10%.

Holiday vouchers in Romania are nominal, non-transferrable, cannot be exchanged for money and are valid for a maximum of 1 year from the date they were issued. Tourism services that can be bought using vouchers may include, for instance, 3 to 5 nights with breakfast or full board for two people at budget hotels (2- and 3-star hotels) from balneary resorts or the Black Sea seaside; 2 nights in luxury hotels (5 stars) for two people, breakfast included; or 2 nights, all inclusive, at 4 star hotels.

By granting holiday vouchers to all the employees in an institution, there is also a social tourism component, because it facilitates the access to a holiday of those who have never had such an experience before.

The manner in which the holiday voucher scheme works and the concepts approached are graphically represented in Figure 1.

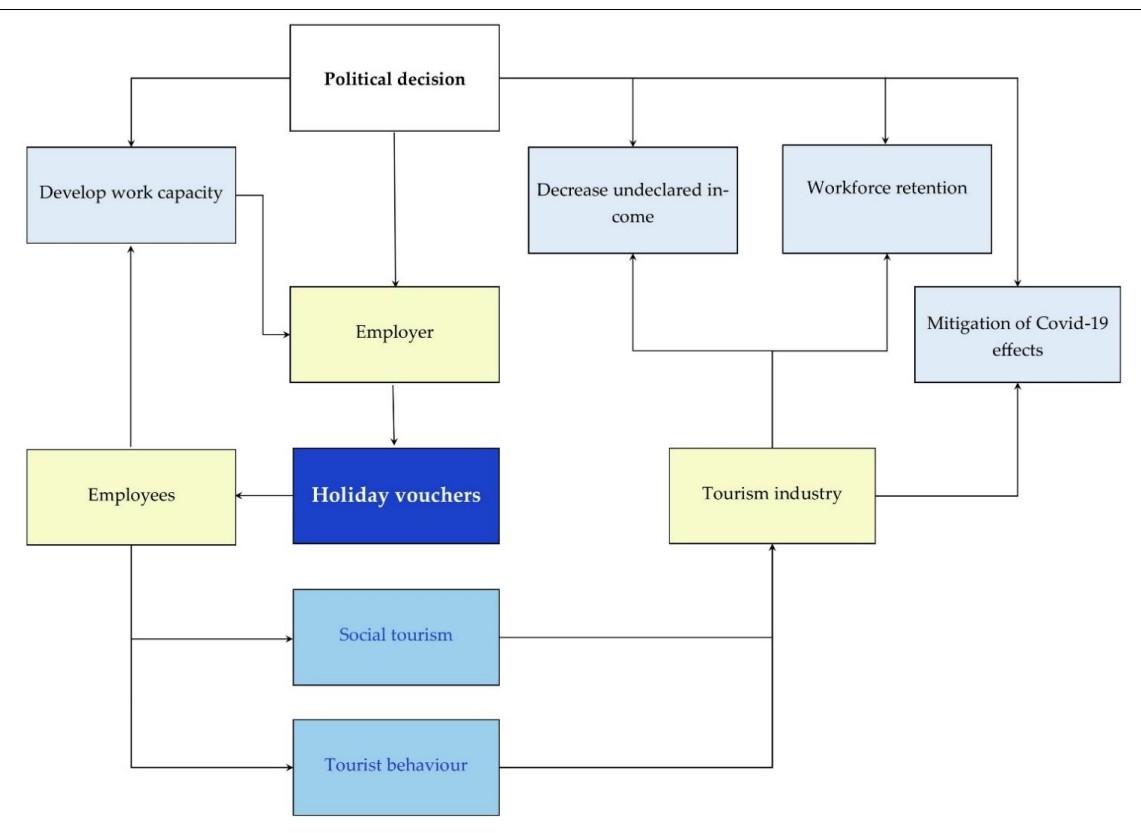

**Figure 1.** Conceptual framework and research context.

## 4. Research Questions

Given the exploratory nature of the study, we formed no hypothesis a priori. However, the questions that guided the study and which we aimed at gaining an answer to are the following:

(1)  Q1. Do the travel agencies' employees consider this payment method useful?
(2)  Q2. Was it a helpful tool in the recovery of the industry after the COVID-19 pandemic?
(3)  Q3. To what extent can the holiday voucher scheme in Romania be seen as successful from the viewpoint of social tourism?
(4)  Q4. How did the vouchers influence tourists' behaviour?

## 5. Research Design

### 5.1. Data Collection

We used a naturalistic inquiry [96], combining focus groups with semi-structured interviews. The focus groups with travel agency workers were used for initial data collection in April 2022, aiming to shed some light on how tourists use holiday vouchers and make travel decisions and if they generally need any advice on the part of the travel agents.

For the interviews, we chose a non-random sampling technique, purposive sampling [97,98], using maximum variation sampling [99]. For interviewing tourists who used holiday vouchers, we selected one travel agency from Craiova, the interviewees being public sector employees who had bought holiday packages from the travel agency at least once during the past 3 years. Interviews were carried out during the spring, summer and December of 2022 at a travel agency in Craiova, Romania. Interviewees were approached by one of the authors and agreed to take part in the survey. Most of the interviews took place in the travel agency, mainly when tourists came to the agency to either pay for the remaining amount of money for the holiday they bought or to pick up their travel documents. As a few respondents could not stay for a face-to-face interview, we carried out some of them over the phone. Every interview took between 10 and 25 min.

There were 70 people who agreed to share their opinions and travel history with us. The characteristics of the sample are given in Table 1 and the research process is presented in Figure 2.

**Table 1.** Sample profile (percentages).

| Age (Y.O.) | | | | Gender | | Family Circumstances | | | | Occupation | |
|---|---|---|---|---|---|---|---|---|---|---|---|
| 18–34 | 35–44 | 45–54 | Over 55 | M | F | Living Alone | Single Parent | In a Relationship, No Children | In a Relation with Children | Blue Collar Jobs | White Collar Jobs |
| 11.43 | 34.29 | 22.86 | 31.43 | 38.57 | 61.43 | 10.00 | 7.14 | 21.43 | 61.43 | 42.86 | 57.14 |

### 5.2. Semi-Structured Interviews

Before launching the survey, a thorough review of social tourism research and mainly the initial focus group helped us shape the questions for the semi-structured interviews to a large extent. Respondents were interviewed based on a semi-structured frame of topics [98], including behaviour prior to tourism vouchers, travel characteristics for trips using the vouchers, perceptions of the usefulness of vouchers, as well as demographic data. General demographic information about the participants include gender, age, family role (single/spouse/son/other), family circumstances (living alone/single parent/in a relationship with no children/with children), education (lower/compulsory education/higher/post compulsory) and occupation (blue/white collar jobs). The main purpose of the interviews was the discovery of the informants' feelings, perceptions and thoughts [100,101] on the usefulness of holiday vouchers granted to the Romanian employees.

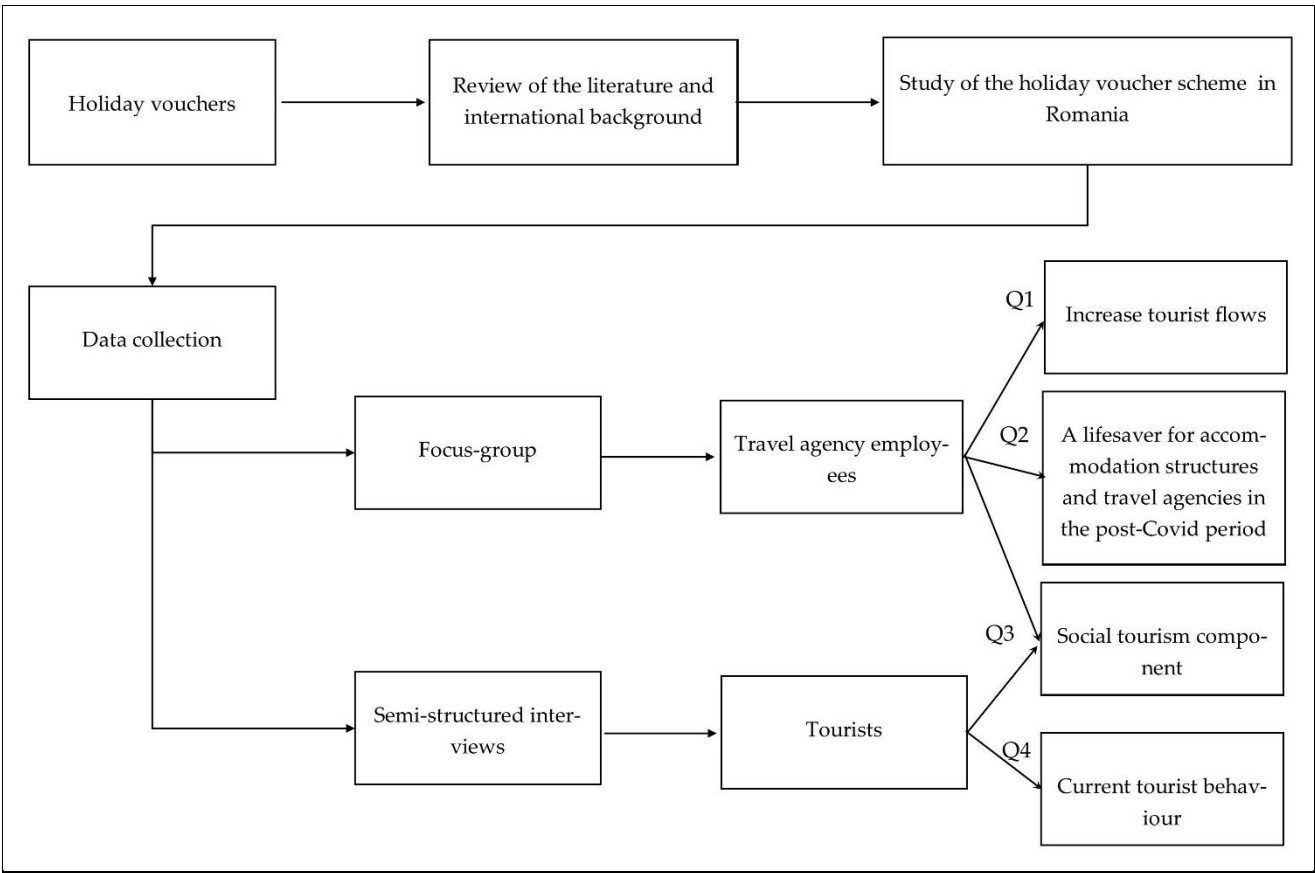

**Figure 2.** Methodological framework.

*5.3. Data Processing*

The interviews were recorded with the consent of the respondents, partially transcribed (as to provide verbatim quotes) and then thematically coded considering the regularly articulated opinions. Thematic analysis was preferred as it offers the necessary flexibility required by the study [45,102]. The main themes were travel behaviour prior to holiday vouchers, travel characteristics for trips using the vouchers and perceptions on the usefulness of vouchers.

To establish credibility of the data and interpretation, we applied the techniques indicated by Lincoln and Guba [103], with a particular focus on peer debriefing and member checks.

## 6. Results and Discussion

*6.1. Focus Groups—Preliminary Findings*

There were two focus groups with workers from travel agencies from Oltenia Region, which is known at a national level for being one of the less developed regions in the country [104]. The first focus group included employees from four small travel agencies owned by local companies (which we called local travel agencies—LTAs) from Dolj, Gorj and Valcea counties, while the second one included workers from a travel agency that has numerous selling offices in Romania (referred to as a nationwide travel agency—NTA). All the participants were employed in agencies that accepted holiday vouchers as a payment method.

From the very beginning, it was obvious that there was a somewhat different perception and different requests received by the travel agents from LTAs and NTAs. Thus, LTA agents pointed to numerous requests for budget holidays, while NTA was selling mostly holidays abroad or very expensive ones. As the vouchers amount to less than 300 EUR, NTA tourists found that it was not enough and they could not buy a holiday for only that amount of money. They generally used vouchers to cover the advance payment for a

holiday in Romania, paying the remaining amount in cash. NTA agents' suggestion was to increase the value of holiday vouchers. When asked *"How do you reckon tourists would plan a future holiday if they no longer benefitted from the vouchers?"*, the answer was unanimous: most of the tourists would probably go abroad if not for the holiday vouchers.

On the other hand, the other focus group with LTA workers, especially those from small agencies with less than five employees, stressed the opportunity provided by holiday vouchers to tourists who had never travelled before. As one of the travel agents said:

*"Among the tourists that bought their vacation from our agency, there was a family of workers that wanted to use the vouchers that both spouses had received to benefit from a couple of days at a hotel offering all-inclusive services; they wished to have a glimpse of the holidays that rich people can afford. Before, they only went camping".* (travel agent, Dolj County)

*"We had a family with 2 children, aged 23 and 19 years old, who bought their first holiday at the seaside. I was stunned to see there were people in their twenties as well as older adults that had never seen the sea!"* (travel agent, Gorj County)

At the same time, they admitted that they were frequently asked by people, especially those with limited financial means, if they could exchange the vouchers for money or give them to relatives and friends, since the holders were not used to travelling or thought that going on holiday would incur extra costs they could not afford.

Although holiday vouchers mean extra reimbursement charges for the travel agencies and a limitation of the margin, most of the agents we talked to agreed that the voucher scheme is beneficial, ensuring continuous selling throughout the year. However, there were also agents who were rather focused on the stipulations implied by the use of the vouchers and would consider it best for employees to be granted a holiday bonus instead of vouchers.

All tourism agents from the LTA group also stressed the undisputable part played by the holiday vouchers for the tourism industry during the pandemic, considering that some of the small travel agencies would have gone bankrupt if not for this alternative means of payment, subsidised by public institutions and which encouraged tourists to travel during that period.

### 6.2. Tourists' Perception

The tourists we interviewed have different social and economic profiles, working different jobs, varying from caretaker or janitor to middle and top management of various institutions, as shown in Table 1.

An overwhelming share of the interviewees (97%) saw the vouchers as beneficial. When asked about the travel behaviour prior to vouchers, 15.71% of the respondents admitted to having never been on a holiday before (Figure 3). They might have taken day trips, but not an entire vacation. This travel inexperience (less than 10 domestic holidays) [105] is of particular importance from a travel decision perspective, as it can be a cause of anxiety and stress [41].

The explanations for the lack of experience regarding travelling and holidays had a somewhat common denominator: lack of money, but also some anxiety about inherent issues such as where to eat during the holiday, if they could find some supermarkets near the hotel to buy cheaper products so as not to spend too much on restaurants, what should they pack, how to get to the hotel, etc. These first-time holiday-goers are people who were not jobless for a long period of time; in fact, most of them had been employed their entire life, but had poorly paid jobs and had to skimp on money so they could cover their basic needs.

*"If I had received some money instead of vouchers, then I would have certainly bought something for my home, such as a gas bottle; but since I received the vouchers, I went for the first time on a holiday".* (personal assistant for disabled person, rural area Gorj County)

*"It is true that we had to pay that 10% tax for the vouchers, but otherwise I would have spent the entire leave working at home, I wouldn't have gone on holiday".* (school janitor, Dolj County)

*"Receiving the vouchers was nothing short of a miracle for us. We are over 50 years old and had never went on vacation. When we got them, the first year we went to the seaside, and the following years we chose some balneary resorts for therapy"*. (driver, rural area, Gorj County)

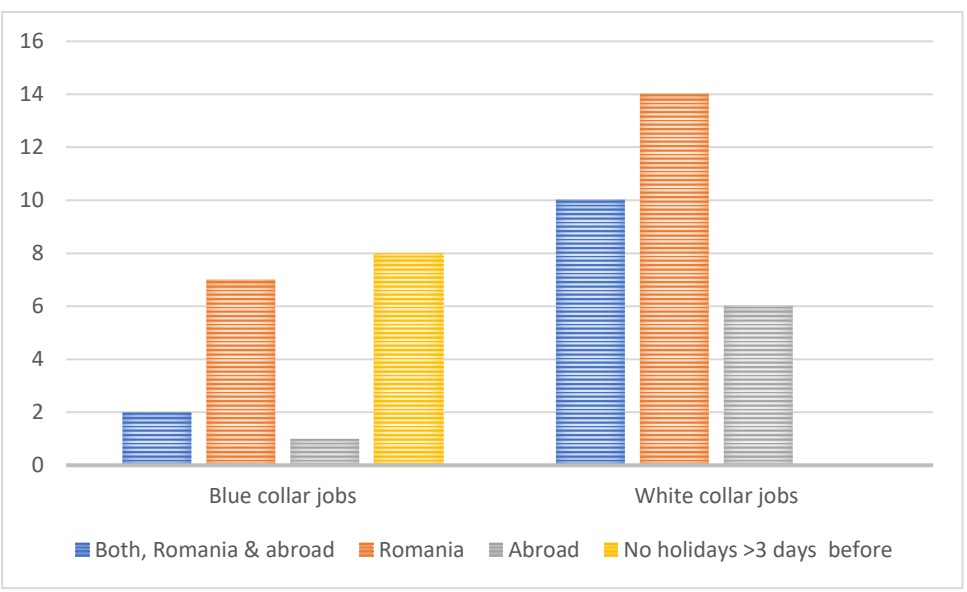

**Figure 3.** The behaviour of the interviewees before holiday vouchers.

Some of the answers shed some light on other difficult social issues, such as feeling marginalised or being ill, as the following people indicated:

*"We have 3 children, I get the minimum wage and my husband is a seasonal worker abroad; there is no way that we could have afforded to go on vacation. If not for the vouchers, we wouldn't have gone to the seaside. But at school, my oldest daughter's teachers always ask her and her colleagues where they spent their holidays. She will feel marginalized eventually if she must always say that she stays at home or only goes to the grandparents, when her colleagues go to a different region every time"*. (social worker, rural area, Gorj County)

*"We are old and sick. When you are 60 and both you and your spouse have minimum wage, and still have parents that are even older and more sick, you must be very careful how you spend the money. You can't afford to even think about travelling. With what money? The little we have we must keep for medicine, God forbids someone falls sick!"*. (Janitor, high school, Dolj County)

There were also people who struggled financially and considered vacation just a prolonged weekend, so as to offer their children some opportunity for seeing other places:

*"I don't earn too much, and being a single mother, there really was no extra money to pay for a vacation. I was struggling to save up the spending money for one, maybe two week-ends per year, somewhere not too far, so as to see at least the main attractions in the neighbourhood counties. But now, we can go farther and see well-known destinations in Romania"*. (51 years old, single mother)

Regarding the much more experienced respondents, 41.43% used to go on holiday in Romania before receiving the vouchers, 28.57% went alternatively in Romania and abroad and another 14.29% used to spend their holidays abroad.

We asked those who chose Romania even before the vouchers how their pattern changed once they received the holiday vouchers. Most of them increased the frequency of holidays, but there were also tourists who extended their stay:

*"We had more vacation days . . . we went not only to the seaside, as usual . . . now we also added some days somewhere in the mountain area"*

*"We took several holidays, and we also chose a better hotel and we paid for the meals beforehand"*

*"Before, I could only seldom afford to go on a holiday; but ever since I received the vouchers, I was more tempted to make some plans and go on vacation"*

*"Thus we could afford a second holiday in a year"*

No matter the background, people with blue collar jobs and those from middle and top management, familiar with luxury, perceive the vouchers as having a positive influence, although by far, the greatest impact was for the people with limited financial means, as some of the tourists testified:

*"The vouchers were a godsend! Me and my husband are sick, so we must go for treatment at a balneary resort. We should have gone much sooner, but as we didn't have the money, we couldn't . . . Now, we still have to pay something, but it's completely different . . . It's doable."*

*"Due to vouchers, we could afford to offer our son the best birthday gift: a trip to Dinoland and Adventure Park near Brasov! All his mates had visited it more than once and he longed to go there, to experience it. Without the vouchers, we certainly wouldn't have paid that much money."*

Some of the interviewees emphasised that even if they usually went on holiday for a week or two, once they had the vouchers, their options improved considerably:

*"We used the vouchers for an all-inclusive trip to the Danube Delta. We had been there before, several times during the last 2 decades, but every time we went camping on a wild beach... Which was very nice, don't get me wrong. But now, we not only had accommodation in one of those cosy, traditional but rather expensive houses, but also enjoyed a boat trip along the channels and lakes. We saw such marvellous things! It was a completely different experience!"* (laboratory technician, Craiova)

The same is true for some of the respondents who usually went on holiday mainly out of necessity, having young children with health problems, for whom the doctor recommended various forms of therapy in balneo spas.

*"We have to go to a salt mine for saline therapy as both our children have respiratory diseases. It wasn't exactly cheap for an entire week to pay for accommodation and treatment. So, as not to add to the cost too much, we always chose the nearest balneary resort in Valcea. But now, due to the vouchers, we can travel further and also see new places, which is great. It doesn't feel any longer that we travel because we are sick, but mainly to see new places, while still minding our health!"* (firefighter, Craiova)

*"Once I had the vouchers, we could travel somewhere just for fun, not out of necessity (treatment for my son)".* (secretary, Segarcea)

For the more experienced domestic travellers, the vouchers also allowed them to pay for better accommodation, with most of them targeting four or five star hotels, and extra services offered such as spa treatments, pools, half-board or packages that include transport to the destination, as the following answers indicate:

*"I wouldn't have stayed at a 4 star hotel otherwise . . . I find the prices quite high, and if I had to pay only cash for the entire stay, I would have chosen a cheaper hotel . . . that's for sure!"* (teacher, Craiova)

*"This provided me the opportunity to have some quality girl time with my teenage daughter: we went to a 4 star hotel just for their spa. We definitely loved being pampered, but if I had to front the money from my pocket for it, I wouldn't have spent it".* (nurse, Craiova)

Other changes compared to the previous period and mentioned by tourists included either buying the package from a travel agency or having hotels booked in advance; before vouchers, they used to go directly to the destination and they would stay at private houses, where they rented a cheaper room and negotiated the price with the owner of the house.

For more than two thirds of the interviewed tourists, vouchers led them to either explore other destinations in Romania or go on their first holiday. The newly chosen destinations were generally located in other parts of the country, such as Bucovina, Maramures or Transylvania, which are the best known Romanian destinations for cultural tourism, or the Romanian seaside and balneary resorts in Oltenia for those having their first holiday.

Even the remaining 14.29% of the respondents who used to spend their holiday abroad largely considered the vouchers as being beneficial both to users and the tourism industry in general: *"yes, definitely, it is good to keep them … thus, tourism is developing … we visit new places in Romania; otherwise, we would be set on travelling only abroad"*.

*"Yes, due to the vouchers, hotels had a continuous flow of tourists. The result was an increase in the quality of services in Romania".*

Still, there were some respondents who pointed out a side effect: they considered that due to holiday vouchers, the costs of accommodation and catering increased. Their main arguments were that travel agencies and tourism facilities accepting vouchers had an extra cost, covered also by tourist, as well as higher demand for holidays in the country due to these vouchers. However, tourism agencies could not charge extra, at least not legally. On the contrary, the larger tourism agencies usually had more than a 10% margin, but had to cut it when they began to accept vouchers since the law did not allow for more than 10%. As for the accommodation facilities, there were no stipulations regarding the margin, only that the price must be the same, no matter the payment method (vouchers or money). The same is true for travel agencies.

We also asked the respondents if they would consider it necessary to differentiate voucher holders based on some criteria, but there was an unanimous answer: no. Some of them provided an explanation for their answer: *"deciding between employees would be subjective"*; *"if they were granted only to the employees with lower wages, they might feel marginalized"*.

When asked *"how would you plan your next holiday if you no longer received the vouchers?"*, some of the respondents did not see any changes regarding the main annual holiday, as they used the vouchers for an extra one, which they would no longer take. For other tourists, it would mean a cheaper holiday, while others (28%) think they would consider a holiday abroad and another 12% most probably would not go on holiday at all the following year.

Regarding the changes they would suggest for voucher legislation, most of them are satisfied with the current form; there were few people suggesting it might be good to extend their validity period.

The tourists interviewed in December 2022 were also inquired about an additional aspect, namely whether they would consider using their holiday vouchers for the Christmas or New Year vacation or for periods with standard rates (during special events such as the Winter Holidays or Easter, package rates are much higher). The majority of tourists who paid for their vacation with holiday vouchers in December opted for vacations outside the holidays, i.e., spring or summer 2023, benefiting from early booking discounts.

*6.3. Discussions*

Q1. The first question we tried to answer was whether employees from travel agencies consider holiday vouchers as a useful payment method. Most of the travel agents involved in the focus groups noted an improvement in the flow of activity generated by the existence of the holiday vouchers on the market and considered them a useful tool for the industry. Although some conditions must be met in order to be able to accept holiday vouchers as a means of payment, the majority of the surveyed workers consider the system beneficial and will continue to accept holiday vouchers for payment.

Q2. Holiday vouchers are used by employees either by purchasing services from travel agencies or directly from accommodation structures. The travel agents agreed that this

means of payment contributed to the survival of small travel agencies in the post-COVID-19 period and implicitly helped the accommodation structures as well.

We can safely assume that if before the pandemic period, vouchers were a booster for the Romanian tourism, then during the pandemic, post-pandemic period and the Russian–Ukrainian war, vouchers have been a lifesaver for accommodation units as well as travel agencies.

Q3. The third key question that guided our study focused on the link between holiday vouchers and social tourism in Romania. The results of the focus groups and interviews clearly indicate that Romanian holiday vouchers are an important tool for social tourism, even if they were not targeting this particular aspect. The employees with minimum wage, which account for 30% of the working force in Romania [106], could no longer afford a holiday if they no longer received vouchers. Nevertheless, it is true that most of the people seeking to exchange them for cash, even for a lesser amount than their actual value, were poorly paid people. Still, the government acknowledged the fact that holiday vouchers are products with high fiscal risk, and as of 1 April 2022, there is a new regulation, according to which all invoices that are totally or partially paid using holiday vouchers must be electronically registered in the database of the General Direction of Public Finances. Thus, the continuous monitoring of the payments using vouchers by the authorities should dissuade any misuse (exchanging them for cash, accepting vouchers for other services than those approved by law or for other people).

The holiday voucher scheme would be an even more comprehensive tool for social tourism if they were granted to a greater extent by private companies as well (being an optional benefit granted to employees, it is up to the employer whether to include this cost in the company's budget, and until now, most of the vouchers granted have been issued by public companies).

Q4. Regarding the tourist's behaviour in choosing the vacation, this facility definitely influences the decision. For tourists spending their holidays in Romania, the services included in the vacation have now been supplemented (either the vacation is longer, or with a second vacation, additional meal or superior comfort) by means of holiday vouchers.

Although white collar job employees mostly appreciated the receipt of holiday vouchers, for blue collar job employees, this benefit is much more important. The holiday vouchers offer them an experience that they would otherwise not be able to afford.

The tourists who had a low budget for the vacation before receiving the vouchers preferred to go on holiday without a prior reservation, finding alternative accommodation at private houses on their arrival day. Granting the vouchers has considerably changed the behaviour of these tourists, as they were obliged to choose authorised and certified accommodation units, since only these units were allowed to accept this method of payment. Moreover, many accommodation units were forced to legalise their activity in order to capitalise on the tourist flows.

As the discussions with tourists indicated, approximately one-third of the interviewees would choose a vacation abroad if not for the holiday vouchers.

Implications for Destination Managers

The diversification of holiday destinations, especially for the tourists who had a second holiday/year due to the vouchers, is an important factor contributing to the development of local resorts. If the budget allows them only one holiday, traditionally, Romanians with an average income stay on the Romanian seaside during July or August. With an extra income, available only for holidays, many Romanians were tempted to explore other less congested areas, such as Transalpina (a high-altitude road crossing the Southern Carpathians) or local tourist resorts within rural areas (e.g., Polovragi, Albac, Arieseni, etc.), as well as to extend their stay from 1 week or a weekend to more than 3 days. Consequently, the occupancy rate of the accommodation units during the week increased.

We must also point out that even if vouchers can be used only for packages that include at least accommodation for 1 night, and thus they are largely used for paying the

accommodation and sometimes meals if the units also have a restaurant, there is no denying the multiplying effect of tourism. A higher number of tourists in a destination leads to higher spending and thus an increase in the selling of other catering and recreational facilities in a particular destination. An increase in the number of tourists should also have positive outcomes for the tourism attractions in the destination proper or along the main access roads. If tourist flows are to be registered throughout the entire year, then there should be a steady influx of money which would ensure the resources necessary to keep the facilities in good repair, increase the quality of the services they offer and thus increase the general competitiveness of the Romanian tourism sector.

Furthermore, for the tourists accustomed to vacation in luxury hotels, vouchers do not mean an important amount of money to be spent, but still, granting vouchers also for this category of employees, belonging mainly to middle and top management, leads to a sort of equilibrium regarding the use of all the accommodation facilities, since these tourists target only 4 and 5 star hotels. If they no longer received vouchers, then they would choose a destination in another country; thus, the Romanian tourism industry would lose an important market segment. Enticing high-spending tourists to Romanian destinations is also important due to the extra budget they have for the holiday, not only for accommodation but also catering and entertainment.

## 7. Conclusions

Romania introduced the system of holiday vouchers beginning in 2015, and several years later, it became common practice among most public institutions. By granting these vouchers, the government has tried to steer citizens from using unregistered accommodation units and reduce the size of undeclared work, thus supporting the tourism economy on the whole. Although they were granted to employees, mostly in the public sector, and suggested to have positive effects on the workforce, it was nevertheless tailored as an instrument to boost domestic tourism.

The emerging research on holiday vouchers has hinted at the economic benefits within the tourism industry, and this study confirms the positive impact on the hospitality sector. The findings point to the general positive perception of holiday vouchers by the employees, who consider them a useful supplementary benefit as well as a necessary one. The only suggestions would be related to a longer period to use them and the need for all private companies to adhere to this practice.

As the current study shows, in Romania, holiday vouchers have led to changes in the behaviour of tourists who used to travel domestically, changes related mainly to a higher number of trips taken, an increase in the nights spent on holiday and sometimes an upgrading of the accommodation facility. Tourists who favoured only holidays abroad before vouchers will largely resume their preferences if they no longer receive vouchers. This would cause a lower return and influx of money for the tourism industry in Romania, having direct consequences such as fewer jobs, which will primarily affect vulnerable categories, namely HORECA employees.

The focus groups with employees from travel agencies as well as the interviews with tourists who used holiday vouchers revealed an even more important side effect: it offered employees with a low income the chance to travel, and for some of them offered access to their first leisure trip ever. Even if the scheme of holiday vouchers in Romania is not intended to have a direct social component, it clearly shows a social impact especially for employees with low wages. Most of the people with white collar jobs could afford to travel anyway; money was not the issue. Vouchers only pushed them to choose to take one trip within the country. However, for the people with blue collar jobs and minimum wage, vouchers have offered them access to one trip per year, and in some cases, their first holiday ever. As Higgins-Desbiolles [39] pointed out, we must not forget that in the era of neo-liberalism, tourism's purpose is to serve human needs and not only deliver profits to the business sector or boost economic growth.

The current study merely explored a small aspect of holiday vouchers in Romania, focusing mainly on the perspective of those who were granted such vouchers. More systematic research is needed on the benefits of holiday vouchers from the perspective of accommodation units, as well as from the point of view of employers who actually bear the expense of the vouchers granted to all employees, so that comparative research can be undertaken. Such an endeavour could contribute to better policies aimed at fostering economic development and increasing the welfare of the employees.

Regarding the limitations of the study, it is acknowledged that interviewing only clients from a single travel agency could bias the sample and impinge the generalisability of the findings. However, representativeness was not required since we were looking for a proof of principle [46], this small-scale qualitative research being aimed at checking for empirical support for the questions of the study.

**Author Contributions:** Conceptualisation, C.D.A. and L.S.P.; methodology, L.S.P.; validation, C.D.A. and L.S.P.; formal analysis, L.S.P.; resources, C.D.A. and L.S.P.; data curation, L.S.P. and C.D.A.; writing—original draft preparation, C.D.A.; writing—review and editing, L.S.P.; supervision, L.S.P. All authors have read and agreed to the published version of the manuscript.

**Funding:** This research received no external funding.

**Informed Consent Statement:** Not applicable.

**Data Availability Statement:** Not applicable.

**Conflicts of Interest:** The authors declare no conflict of interest.

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
