# Peer review of "Romanian Holiday Vouchers: A Chance to Travel for Low-Income Employees or an Instrument to Boost the Tourism Industry?"

_sustainability, doi:10.3390/su15021330_

Round 1

Reviewer 1 Report

The detail review report is attached herewith

Author Response

Dear Reviewer,

We would like to thank you for the time and effort dedicated to providing feedback on our manuscript, as well as for the insightful comments. We appreciate the opportunity to resubmit a revised copy of this manuscript “Romanian holiday vouchers: a chance to travel for low income employees or an instrument to boost the tourism industry?”. We believe your suggestions considerably improved our revised manuscript. Please find attached the reviewer comments with our responses in italics, including how and where the text was modified.

We very much hope the revised manuscript is in line with your recommendation and will be accepted for publication in Sustainability.

Reviewer 2 Report

Thank you for the opportunity to review the article “Romanian holiday vouchers: a chance to travel for low income employees or an instrument to boost the tourism industry?”. The paper addresses an interesting and well researched theme in the recent period about the implications of government actions in supporting different sectors and employees to have access to holidays and to support the tourism sector. The paper is also in line with the section “Tourism, Culture, and Heritage” and with the special issue was submitted to: “Tourism Research and Regional Sciences”.

This study represents a solid effort in the field approached. It is constructed in a mature manner, following the publication standards of the journal, discussing the subject that needs to be comprehensively analyzed because “the study analyzes the influence of the Romanian holiday vouchers scheme on the tourist behavior of employees in order to establish the holiday vouchers` usefulness, their use as a tool in social tourism or for the development of the tourism industry.”, as the authors underline.

Also, the study is written in an adequate manner, with a specific review of the literature and robust research design. The results are presented clearly and coherently, using visuals and text. The figures presented in the paper were relevant to explore the results of the research and the ways these were adapted to the explanations in the text.

As the authors say, one conclusion assumes that “this study adds to the understanding of both the social benefits provided by holiday vouchers and the impact on the hospitality sector” but also inform the readers about implications, limitations of the study.

Moreover, there are some point-by-point observations that should be addressed in this revision.

-          Line 140: the idea needs to be referenced “Moreover, using qualitative data, Kakoudakis et.al. highlighted…”

-          Line 208 “1450 Ron/ year, approximately 300 EUR”, use the term RON; use the same model at Line 98: “EUR 275 per calendar year…” replace with “275 EUR per calendar year”

-          Line 307 and similar entries for English “Dolj county” replace with “Dolj County”

-          Line 468 “inter-views” replace with “interviews”

-          It is not clear why the issue or the influence of COVID pandemic was explained and introduced for the reader (Line 102-107), as this topic is not part of the research questions, or the answers provided.

-          The question in the title should be more clearly presented with an answer in the Conclusions section.

-          Future directions of the research should be added and also to highlight the novelty factor better.

Author Response

(The authors gave the same response as above.)

Reviewer 3 Report

The researchers must also mention other tourist places and how they are using vouchers or any other promotional tools to attract tourists or support the tourism industry. If the Pandemic is the point of reference, authors must show how other tourism destinations have started recovery measures.

Author Response

Dear Reviewer,

We would like to thank you for the time and effort dedicated to providing feedback on our manuscript, as well as for the insightful comments. We appreciate the opportunity to resubmit a revised copy of this manuscript “Romanian holiday vouchers: a chance to travel for low income employees or an instrument to boost the tourism industry?”. We believe your suggestions considerably improved our revised manuscript.

We very much hope the revised manuscript is in line with your recommendation and will be accepted for publication in Sustainability.

Round 2

Reviewer 1 Report

Thank you very much for addressing the comments raised during the review process.

Reviewer 2 Report

Thank you for the opportunity to review the revised version of the paper “Romanian holiday vouchers: a chance to travel for low income employees or an instrument to boost the tourism industry?”.   

The authors responded and explained with reasonable arguments and corrected all the remarks and observations highlighted in the previous review and the results suggest a more consistent and logical text.

To sum it up, the authors developed a more in-depth theoretical presentation about the subject, integrating the suggested aspects of the review.

I consider that the paper is publishable after a final check from the authors.

Reviewer 3 Report

The authors have extensively modified the manuscript.